# Federated Generalized Novel Category Discovery with Prompts Tuning

**Lei Shen**
*TMLR Group, Department of Computer Science*
*Hong Kong Baptist University*
*cslshen@comp.hkbu.edu.hk*

**Nan Pu**
*Department of Information Engineering and Computer Science*
*University of Trento*
*nan.pu@unitn.it*

**Zhun Zhong**
*School of Computer and Information*
*Hefei University of Technology*
*zhunzhong007@gmail.com*

**Mingming Gong**
*School of Mathematics and Statistics*
*University of Melbourne*
*mingming.gong@unimelb.edu.au*

**Dianhai Yu***
*Baidu Inc.*
*yudianhai@baidu.com*

**Chengqi Zhang**
*Department of Data Science and Artificial Intelligence*
*Hong Kong Polytechnic University*
*chengqi.zhang@polyu.edu.hk*

**Bo Han***
*TMLR Group, Department of Computer Science*
*Hong Kong Baptist University*
*bhanml@comp.hkbu.edu.hk*

**Reviewed on OpenReview:** *https://openreview.net/forum?id=dVMESwnMlo*

## Abstract

Generalized category discovery (GCD) is proposed to handle categories from unseen labels during the inference stage by clustering them. Most works in GCD provide solutions for unseen classes in data-centralized settings. However, unlabeled categories possessed by clients, which are common in real-world federated learning (FL), have been largely ignored and degraded the performance of classic FL algorithms. To demonstrate and mitigate the harmful effect of unseen classes, we dive into a GCD problem setting applicable for FL named FedGCD, analyze overfitting problem in FedGCD in detail, establish a strong baseline constructed with state-of-the-art GCD algorithm simGCD, and design a learning framework with prompt tuning to tackle both the overfitting and communication burden problems in FedGCD. In our methods, clients first separately carry out prompt learning on local data. Then, we aggregate the prompts from all clients as the global prompt to help capture global knowledge and then send the global prompts to local clients to allow

---

*Correspondence to Dianhai Yu <yudianhai@baidu.com> and Bo Han <bhanml@comp.hkbu.edu.hk>

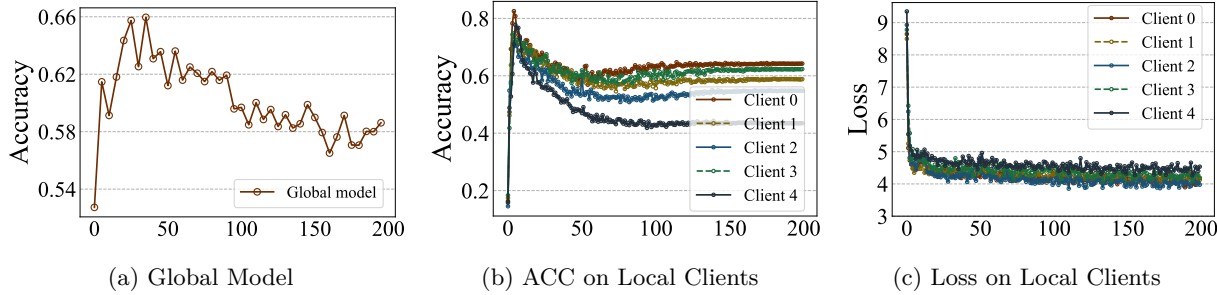

Figure 1: Overfitting of fine-tuning the last block with simGCD on Cifar-100. Horizontal axis represents global training epoch in Federated Learning. Subfigure (a) shows the accuracy of global model during training process. Subfigure (b) and (c) report accuracy and loss on local clients, respectively.

access to broader knowledge from other clients. By this method, we significantly reduce the parameters needed to upload in FedGCD, which is a common obstacle in the real application of most FL algorithms. We conduct experiments on both generic and fine-grained datasets like CIFAR-100 and CUB-200, and show that our method is comparable to the FL version of simGCD and surpasses other baselines with significantly fewer parameters to transmit.

# 1 Introduction

With advanced supervised learning techniques which have been greatly developed in recent years, machines can approach human performance in some types of tasks such as classification and segmentation. However, this enormous ability of machines relies heavily on the annotations given by people and requires a large amount of labeled data, which may be deficient in many real scenarios due to the varying data distributions/tasks and the substantial labor required. Several fields like domain adaptation (DA) and domain generalization (DG) are proposed to tackle the situation where distribution changes, while all these methods fail to solve the problem where the tasks met by the models are varied. On the other hand, cross-task transfer learning Transfer (TL) strategies have gained great success and are widely applied with the labels of new tasks (Vaze et al., 2022). However, transfer learning cannot help new tasks without the labels from new tasks. To deal with the problem of the lack of labels in new tasks, novel category discovery (NCD) and generalized category discovery (GCD) first learn a feature extractor with semi-supervised representation learning for making full use of both the labeled and unlabeled data. Then, a non-parametric clustering or parametric classification method is applied to cluster the unlabeled data from new tasks (Wen et al., 2023a).

Most previous works in FL assume that the data from clients are fully labeled and all classes in test data appear in training data (McMahan et al., 2017). However, this assumption is becoming harder to maintain for FL systems than in the data-centralized setting. Therefore, considering GCD problems in the FL setting is of great importance for real-world applications. Most previous work in GCD only provides solutions for unseen classes in the data-centralized setting, while similar situations are more common in FL as described in Section 2.4. In the meantime, FL setting brings more challenges to the GCD problem since both representation learning and classification are more demanding to learn with distributed or even heterogeneous data in each client, which is often encountered in FL. We found the overfitting problem of GCD even worse and more complicated in the data distributed scenario, where each client owns significantly less data compared with the centralized setting. In Federated Generalized Category Discovery (FedGCD), overfitting for labeled (old) class leads to performance descent in a novel class. Moreover, we witness a test performance gap between local training and test data due to data deficiency to train a complete large-scale model or even a layer of it, as demonstrated in Figure 1. We empirically show our method can effectively alleviate the overfitting problem in FedGCD by local prompt tuning with less but more effective trainable parameters in Section 5.6.

Moreover, the real application of FL always meets strict restrictions for communication costs. Our prompt tuning paradigm perfectly responds to the requirement of reducing communication costs in the real employment of FL algorithms. Previous FedGCD requires training the whole model from scratch (Zhang et al., 2023a) or

fine-tuning the last blocks in local clients (Zhang et al., 2023a; Pu et al., 2023a). Previous FedGCD methods like FedoSSL and AGCL require the transmission of the whole local models, which typically consist of millions of trainable parameters, together with auxiliary components such as local centroids and Gaussian mixture models. The training paradigm causes a great communication burden for both local clients and the central server. Our method FedGCD-P exploits recent Parameter-Efficient Fine-Tuning (PEFT) methods to tackle this challenging but important problem: how can we effectively learn both local and global presentation without modifying a large number of parameters, which costs unbearable communication bandwidth and demands the local client to own great computation ability? Specifically, we adopt prompt tuning (Jia et al., 2022) to achieve this goal: by only modifying a small number of prompt parameters, we extract useful presentation for local datasets, which significantly reduces communication and computation costs, making our method far more practical and resource-friendly.

We conduct experiments on commonly-used classification datasets such as CIFAR-100 and CUB-200 with a self-supervised pre-trained Vision Transformer DINO-ViT-B/16, demonstrating that our method obtains state-of-the-art performance with only 0.7% parameter transmission.

We summarize our contributions as follows:

- We first discover the overfitting problem of FedGCD, which significantly degrades the performance of previous FedGCD methods and causes heavy system overhead in federated systems, thus is of great significance for improving FedGCD methods especially in the time of large models.

- We propose FedGCD-P, a FedGCD method based on prompt tuning to handle the overfitting problem of FedGCD, and mitigate the communication burdens in generic FL settings.

- We empirically demonstrate the effectiveness of our method to be comparable or even outperform baselines with the transmission of significantly fewer parameters by solving the overfitting problem.

## 2 Related Work

### 2.1 Federated Learning

Federated learning (FL) was first proposed by (McMahan et al., 2017) as a decentralized training paradigm without sharing training data. However, naive FedAvg faced many problems, two severe problems are (1) Data Heterogeneity: In the real application, different distributions always occur in different clients, which violates the basic i.i.d. (independent and identically distributed) assumptions of most machine learning algorithms. Especially when using FedAvg, heterogeneous data will cause severe client drift. To solve this problem, plenty of works have been proposed to restrict local updates either with a regularization term (Acar et al., 2020; Li et al., 2021a) or by correcting the local update (Karimireddy et al., 2020) to prevent them from diverge too much from global model; (2) Communication cost: FedAvg ask the full transmission of the whole model or gradients of it, which is of thousands of parameter and non-trivial to communicate. A large number of parameter decoupling methods (Arivazhagan et al., 2019; Collins et al., 2021; Chen & Chao, 2021; Jiang & Lin, 2022) have been proposed to update only part of the model to reduce the communication cost of transmitting the model or gradients, while keeping similar or even better results compared to update the whole model. However, due to communication and computation resource limitations, large-scale vision Transformers (ViT) are explored in FL with a relatively low frequency. A few previous works (Che et al., 2023; Sun et al., 2023; Wei et al., 2024) focused on fully supervised and close-set settings, which is a typical setting in idealistic FL. However, when applying FL in the real world, the fully labeled data and close-set assumption are not applicable, therefore making those works impractical in such a situation.

### 2.2 Federated semi-supervised Learning

To tackle the practical issue of full-supervised assumption above, there are already some works attempting to combine unsupervised or semi-supervised together with FL, to enable real-world FL even when label missing exists in local clients. Some of them (Albaseer et al., 2020; Li et al., 2023) utilize a model trained on labeled

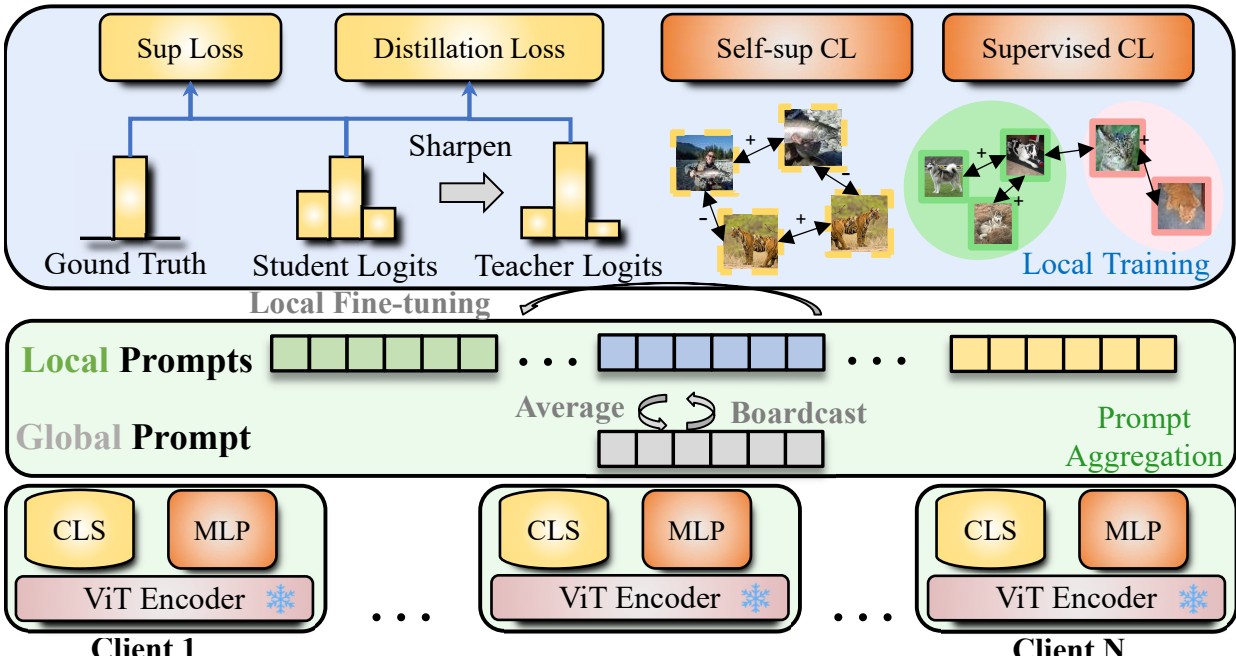

Figure 2: Overall Framework of FedGCD-P. The images with a solid line frame are labeled data, while the images with a dotted line frame are unlabeled data. We fix the encoder of the Vision Transformer. At the beginning of one federated round, Global Prompt serves as the initial value of Local Prompts, while the Local Prompts are obtained from local fine-tuning. While at the end of the federated round, Local Prompts aggregate by average to get Global Prompts.

data to produce pseudo labels to carry out semi-supervised learning in FL. While some previous works (Zhang et al., 2020; Long et al., 2021) use consistency regularization, a semi-supervised technique commonly used, aiming at minimizing the consistency loss between two representations of the same sample. There are also works like FedMatch (Jeong et al., 2020) combining pseudo label and consistency regularization to boost the performance of semi-supervised learning. However, all those works assume a close-set setting, which is not completely practical in FL, e.g. some clients may have not observed data from some classes previously and thus are not possible to correctly label them.

## 2.3 Generalized Category Discovery

Due to the frequent happening of unlabeled class in training data and test data. Novel category discovery (NCD) (Han et al., 2019b) is first proposed for leveraging training data and cluster test data from unlabeled classes more reasonably. Several previous works (Han et al., 2019a; Jia et al., 2021; Zhao & Han, 2021; Zhong et al., 2021a;b; Fini et al., 2021) for NCD also have shown the ability to cluster unknown class data with knowledge learned from unlabeled classes. However, these works are all based on the assumption that all the test data comes from unknown classes, which is impractical in most application scenarios. For example, (Han et al., 2019b) involves a procedure to predict the number of new categories in the unlabeled data. In this process, the estimated number of classes is considered non-overlapped, and overlapped classes in labeled and unlabeled data will bias the prediction of the number of unknown classes. Therefore, generalized category discovery (GCD) has been proposed by (Vaze et al., 2022), which extends the classes of test data to both unknown classes in unlabeled training data and classes already appeared in labeled training data. At the same time, Open-world Semi-supervised Learning (Open-world SSL) (Cao et al., 2022) proposed a very similar problem setting with GCD. Later, DCCL (Pu et al., 2023b) introduce Conceptional Contrastive Learning (CCL) to enhance representation learning by exploiting hierarchical classification information, and their cluster methods follow the semi-supervised cluster paradigm of GCD (Vaze et al., 2022). While simGCD and GPC (Zhao et al., 2023) propose to exploit parametric classification (Wen et al., 2023b) with prototypes

learning, respectively through direct optimization and EM-like optimizing Gaussian Mixture Models (GMM). PromptCAL (Zhang et al., 2023b) and SPTNet separately introduce GCD methods based on soft prompt and spatial prompt tuning to solve GCD problems. However, all the above works discuss the GCD problem in a centralized setting, leaving the GCD problem in the Federated setting an under-discussed field. To the best of our knowledge, only two previous works FedoSSL (Zhang et al., 2023a) and AGCL (Pu et al., 2023a) have discussed the GCD problem in the Federated setting, and our work first discusses the overfitting problem of existing GCD methods and explore prompt learning in the GCD problem in Federated setting.

## 2.4 Federated GCD

In federated learning, data are decentrally scattered among all clients, and it is hard to require all participants to label local data; therefore, a great number of unlabeled data exist in FL systems. Ignoring these unlabeled data will greatly damage the performance of federated models when faced with the same class data as the unlabeled data and misclassify the unlabeled new class data into the labeled classes. To leverage these unlabeled data properly and correctly cluster the new task data, formulating federate generalized category discovery (FedGCD) and designing corresponding generalized category discovery methods to solve the FedGCD problem is necessary and beneficial. However, to the best of our knowledge, there are only two previous works FedoSSL (Zhang et al., 2023a) and FedGCD (Pu et al., 2023a) attempted to solve the GCD problem under the federated setting. Below we discuss them in detail and compare their differences with our work.

**FedoSSL.** In FedoSSL (Zhang et al., 2023a), local feature extractors is trained with unbiased semi-supervised learning loss in local clients and aggregate all local feature extractors to obtain a global one. However, a traditional CNN architecture is adopted by FedoSSL, which may face the problem of not having enough representation ability due to the lack of training data in the federated setting. Different from FedoSSL, to enhance the representation ability of the global federated learning model, we utilize prompt learning with joint training of large-scale pre-trained visual transformer (ViT), aiming at better discovering semantic clusters in the unlabeled data. As for the clustering of unlabeled data, a prototype learning method is exploited to carry out clustering in the server in FedoSSL. The way of clustering with uploading local centroids may be exposed to the risk of privacy leakage, while replacing Sinkhorn-Knopp based clustering with general k-means method may witness a performance drop as stated in FedoSSL (Zhang et al., 2023a). Therefore, with a more representative feature extractor to obtain better semantic features, we choose to take advantage of k-means to cluster the unseen categories without auxiliary information.

**AGCL.** AGCL (Pu et al., 2023a) attempts to share information through the Gaussian mixture model (GMM), however, such methods are faced with certain privacy risks, since a generative model trained with local data may reveal private information about the clients. What's more, AGCL (Pu et al., 2023a) fine-tunes and transfers the last block of the ViT, which costs large communication overhead as Table 2 shown and makes the method unpractical for closed-source pre-trained model following the trend in current Large Language Models, e.g. GPT-4, Claude 3.5.

## 3 Problem Definition

Before introducing the FedGCD setting, we introduce the formulation of the GCD problem (Vaze et al., 2022), FL setting (McMahan et al., 2017; Smith et al., 2017) and present some preliminaries. Finally, we introduce the FedGCD setting defined in (Pu et al., 2023a).

### 3.1 Generalized Category Discovery

Our GCD setting follows (Vaze et al., 2022). Specifically, we assume that the training dataset $D = D_l \cup D_u$ comprises two subsets: a labeled set $D_l = \{x_i, y_i\}_{i=1}^{N_1} \subset X_l \times \mathcal{Y}_l$ with its label space $\mathcal{Y}_l = C_{Seen}$, and an unlabeled set $D_u = \{x_i\}_{i=1}^{N_2} \subset X_u$ with its underlying label space $\mathcal{Y}_u = C = C_{Seen} \cup C_{Unseen}$. Here, $C$, $C_{Seen}$, and $C_{Unseen}$ denote the label set for All, Seen, and Unseen classes, respectively. Following (Vaze et al., 2022), we assume the number of the All classes $|C|$ is known.

### 3.2 Federated Learning

FL algorithms can be split into Generic FL (GFL) and Personalized FL (PFL) according to the place where the inference happens, and they focus on different optimization goals with varied application scenarios. Below we briefly introduce both kinds of FL. In this work, we mainly focus on the Generic FL setting.

#### 3.2.1 Generic FL

The GFL aims to make $N_L$ clients collaboratively learn a global model parameterized as $\theta$ used to conduct prediction on the server. Each client has its local training dataset, we denote the training dataset of client n as $D_n^L$. Thus, the local objective function $\mathcal{L}_n(\theta)$ on client $n$ is also different from client to client. The global optimization object of GFL is defined as (Karimireddy et al., 2020):

$$\min_{\theta \in \mathbb{R}^d} \mathcal{L}_G(\theta) := \sum_{n=1}^{N_L} p_n \mathcal{L}_n(\theta) := \sum_{n=1}^{N_L} p_n \mathbb{E}_{x_n \in D_n^L} \ell(f(\theta, x_n), x_n), \tag{1}$$

where $(x_n, y_n) \in D_n^L$ is the sample from $D_n^L$, $f(\theta, x_n)$ is the prediction of the trained model, $d$ is the number of model parameters, $p_n > 0$ and $\sum_{n=1}^{N_L} p_n = 1$. Usually, $p_m = \frac{N_n^L}{N}$, where $N_n^L$ denotes the number of client $n$'s samples in local training dataset and $N = \sum_{n=1}^{N_L} N_n^L$. The global model refers to the model obtained from optimizing the GFL objective.

#### 3.2.2 Personalized FL

While the object of GFL is to learn a model suitable for all training distribution, the PFL requires less from a single model and chooses to learn multiple personalized models fitting on local datasets separately: (Chen & Chao, 2021; Li et al., 2021b):

$$\min_{\Omega, \theta_1, ..., \theta_{N_L}} \mathcal{L}_P(\Omega, \theta_1, ..., \theta_{N_L}) := \sum_{n=1}^{N_L} p_n \mathbb{E}_{x_n \in D_n^L} \ell(f(\theta_n, x_n), x_n) + \mathcal{R}(\Omega, \theta_1, ..., \theta_{N_L}), \tag{2}$$

where $\Omega$ is the collaboration scheme of clients, $\mathcal{R}$ is the regularizer (Chen & Chao, 2021), both of which vary from algorithm to algorithm, . The obtained personalized models are named as Personalized Models (PMs).

### 3.3 Federated Generalized Category Discovery

From the introduction of problem formulation of NCD and FL, We now introduce the formal formulation of FedGCD (Pu et al., 2023a). We assume there are $N_L$ clients in the FL system, and client n maintains a local model $\theta^n$. A global model is initialized in two ways according to the parameter part, (1) randomly initialized, (2) initialized with pre-trained model (Caron et al., 2021) like in our experiments. After initialization, the global model is distributed to all clients, i.e. $\forall n \in \{0, 1, \ldots, N_L\}, \theta_0^n = \theta_0^G$. Given the local labeled training dataset on n-th client $D_{n,l}^L = \{(x_i, y_i)\}_{i=1}^{N_{n,l}^L} \in X_{n,l}^L \times \mathcal{Y}_L^n$ with the corresponding data set $X_{n,l}^L$ and label set $\mathcal{Y}_n^L$, and the local unlabeled training dataset $D_{n,u}^L = \{(x_i, y_i)\}_{i=1}^{N_{n,u}^L} \in X_{n,u}^L$, client n is supposed to train its local model $\theta^n$. In order to keep the same setting with FedGCD (Pu et al., 2023a), we also assume the label space of each client cannot be the same as or include that of another client, i.e. for client $i$ and $j$, $i \neq j$, $\mathcal{Y}_i^L$ and $\mathcal{Y}_j^L$, $\mathcal{Y}_i^L \cup \mathcal{Y}_j^L \neq \mathcal{Y}_i^L$ or $\mathcal{Y}_j^L$. Following the common FL setting (McMahan et al., 2017), we simulate and control the data and label heterogeneity that often exists in both real-world FL and GCD applications, with the Latent Dirichlet Sampling (Dir) partition method (Hsu et al., 2020). In our setting, we mainly consider the Generic FL setting where a global model is used for inference by all clients in the FL system.

## 4 Methodology

### 4.1 Local Prompt Tuning

From the observation above, we think about what may cause the performance gap between training and test data and the reason why a similar phenomenon hasn't been observed in a centralized setting. The assumption

made by us is that the local clients own significantly fewer data samples in their number compared with centralized settings. Moreover, less diversity due to the notorious problem called data heterogeneity in FL significantly increases the risk of overfitting the training on local clients. To alleviate the effects brought by those problems, we propose to apply prompt tuning for local training. To effectively tune the local prompt, we adopt the loss style from state of art GCD algorithm (Wen et al., 2023b). To further boost prompt tuning on local clients, we also propose a loss applying to the prompt itself. As shown in Figure 2, our overall learning objective is composed of four parts. We first introduce supervised classification loss and self-distillation loss, together with prompt-tuning to FedGCD setting.

**Loss on [CLS] token.** Fine-tuning of Vision Transformers on downstream classification tasks always relies on [CLS] token to utilize supervision information. Therefore, for prompt tuning, we follow previous works to output class logit with a multilayer perceptron (MLP) as classification prediction and pseudo label, and apply cross-entropy loss for comparing model prediction and label together with the pseudo label. Our loss function on MLP consists of two parts: supervised classification loss and self-distillation loss with entropy regularization. The denotation follows those in the Section 3.3. For the supervised part, given $p(x_i)$ is the output logit of MLP for local sample $x_i \in D_n^L$, supervised classification loss is defined as

$$L_{cls}(D_n^L) = \frac{1}{N_{n,l}^L} \sum_{x \in D_{n,l}^L} CE(\text{softmax}(p(x_i)), y_i), \tag{3}$$

, where $y_i$ is the class label of the sample $x_i$. $CE(\cdot, \cdot)$ denotes cross-entropy loss. As for the self-distillation part, it involves cross-entropy of output logit of $x_i$ sharpened with a relatively low temperature $\tau_s$ and logit of different views (random argumentations) $x_i'$ with a sharper temperature $\tau_t$ as the pseudo label.

$$L_{SD}(D_n^L) = \frac{1}{N_n^L} \sum_{\substack{x_i, x_i' \in \text{Aug}(D_n^L) \\ x_i \neq x_i'}} CE(\text{softmax}(p(x_i)/\tau_s), \text{softmax}(p(x_i')/\tau_t)), \tag{4}$$

where $\text{Aug}(D_n^L)$ denote the augmented local training data.

After introducing the loss function associated with the output of MLP, we also apply supervised and self-supervised contrastive loss for better representation ability with prompts. For supervised contrastive loss, class labels are used to identify if two samples are from the same class instead of providing direct classification guidance, and this loss aims to reduce the distance of samples from the same class in feature space and increase that of samples from different classes, the specific form is as follows:

$$L_{sup-con}(D_n^L) = \frac{1}{N_{n,l}^L} \sum_{x_i \in D_{n,l}^L} \sum_{x_j \in N(x_i)} \frac{\exp(z_i, z_j)}{\sum_{x_n \in D_{n,l}^L} \exp(z_i, z_n)} \tag{5}$$

where $N(x_i)$ is the set that all samples own the same label $y_i$ with $x_i$, i.e. $N(x_i) = \{x \in D_n^L | y_j = y_i\}$, $z_i, z_j$ is the feature of $x_i, x_j$. As for the self-supervised contrastive loss, we define it in the following form:

$$L_{self-con}(D_n^L) = \frac{1}{N_n^L} \sum_{x_i \in D_n^L} \sum_{\substack{x_i, x_i' \in \text{Aug}(D_n^L) \\ x_i \neq x_i'}} \frac{\exp(z_i, z_i')}{\sum_{x_n \in D_n^L} \exp(z_i, z_i')} \tag{6}$$

, where $z_i'$ is the feature of augmented image $x_i'$.

With the four parts of loss, we have the total loss function below:

$$L(D_n^L) = \lambda * (L_{cls}(D_n^L) + L_{sup-con}(D_n^L)) + (1 - \lambda) * (L_{SD}(D_n^L) + L_{sup-con}(D_n^L)) \tag{7}$$

where $\lambda$ is a hyper-parameter controlling the importance of supervised and unsupervised learning objectives. On each client participating in current training, the overall learning objective is in the form of equation 7.

**Loss on [PMT] tokens.** After implementing the loss on [CLS] token, we attempted to carry out prompt tuning on local clients, surprisingly finding prompt tuning instead of fine-tuning the whole block effectively relieves the overfitting problem. We assumed that fine-tuning the prompt with only the learning objective of the output of [CLS] token was not sufficient for full utilization of knowledge from local data, especially the labeled data. Motivated by this assumption, we thought the lack of supervision and self-supervision on the output of prompt tokens was the root of the problem. Therefore, we added auxiliary MLP for [PMT] tokens to assist the supervision from labeled samples and the self-supervision from all samples, our training objective for [PMT] tokens follows the above overall learning objective in equation 7. After applying Loss on [PMT] tokens, the total loss turns into:

$$L_{total}(D_n^L) = L(D_n^L) + L_{PMT}(D_n^L) \tag{8}$$

, where $L_{PMT}(D_n^L)$ is the loss applied to the outpt of MLP for [PMT] tokens. However, our empirical results show the learning objective on prompt tokens doesn't significantly improve the representation ability of prompts, which we will elaborate more in Section 5.3.

## 4.2 Global Aggregation

We only aggregate the prompt part of the Vision prompt transformer. In practice, this aggregation scheme greatly reduces the communication burden of both clients and the server, together with the storage burden of servers in FL systems with plenty of clients. Our aggregation scheme for local prompts from participating clients follows the classic Federated algorithm FedAvg (McMahan et al., 2017), which averages the prompt from clients who take part in the current training round by its number of training samples.

$$\theta_{t+1}^G = \sum_{n=1}^{N^L} \frac{N_n^L}{N} \cdot \theta_t^n \tag{9}$$

, where $\theta_{t+1}^G$ is the global model parameters at t+1 round, $\theta_t^n$ is the local model parametes of n-th client at t round. After aggregating the prompt parameters, the server sends the global prompt back to clients, clients update their local prompt with the global prompt. Through this aggregation process, clients benefit from the knowledge of other clients.

## 5 Experiments

Table 1: Comparison of different methods on CIFAR-10, CIFAR-100, CUB-200, and Oxford-Pet datasets. §: As ACGL (Pu et al., 2023a) official implementation hasn't been release, we collect the origin result of ACGL from (Pu et al., 2023a). *: FedoSSL failed to converge on CIFAR-10 and CIFAR-100 in our setting following their hyper-parameters, and we rerun them with modified parameters but still failed.

| Method | CIFAR-10 (%) | | | CIFAR-100 (%) | | | Imagenet-100 (%) | | | CUB-200 (%) | | | Oxford-Pet (%) | | |
|---|---|---|---|---|---|---|---|---|---|---|---|---|---|---|---|
| | All | Seen | Unseen | All | Seen | Unseen | All | Seen | Unseen | All | Seen | Unseen | All | Seen | Unseen |
| 5 clients (full participation) | | | | | | | | | | | | | | | |
| **Fed-GCD** | 93.3 | 95.3 | 91.3 | 63.6 | 63.4 | 63.8 | 77.3 | 84.7 | 70.0 | 49.1 | 52.4 | 45.9 | 84.6 | 82.8 | 86.4 |
| **Fed-SimGCD**§ | 86.4 | 92.3 | 80.4 | 66.0 | 62.1 | 69.8 | 76.3 | 88.7 | 63.8 | 63.0 | 63.4 | 62.5 | 82.6 | 81.9 | 83.3 |
| **FedoSSL*** | 65.6 | 65.5 | 65.7 | 29.6 | 33.2 | 26.0 | 46.6 | 59.1 | 34.0 | 52.0 | 56.5 | 47.7 | 76.0 | 84.5 | 67.0 |
| **AGCL** | 82.5 | 83.4 | 82.2 | 54.2 | 54.6 | 54.0 | 73.1 | 78.1 | 67.0 | 53.1 | 52.9 | 54.2 | 81.4 | 82.0 | 80.7 |
| **FedGCD-P** | 94.1 | 94.0 | 94.1 | 69.3 | 71.3 | 67.2 | 79.0 | 87.3 | 70.8 | 61.3 | 63.5 | 59.2 | 83.1 | 85.4 | 80.7 |

Table 2: # parameters needed to transmit versus methods on CIFAR-100.

| Method | Fed-GCD | Fed-SimGCD | FedoSSL | AGCL | FedGCD-P |
|---|---|---|---|---|---|
| # Parameters | 7.1M | 7.1M | 7.9M | 8.7M | 55.3k |

## 5.1 Experimental Setup

**Dataset.** To fairly evaluate the performance of FedGCD methods, we conduct a comparison for all methods on two commonly-used generic image classification datasets (i.e., CIFAR-10 (Krizhevsky et al., 2009), CIFAR-100 (Krizhevsky et al., 2009)) and two fine-grained image classification datasets (i.e., CUB-200 (Welinder et al., 2010), Oxford-Pet (Parkhi et al., 2012). We leverage the Latent Dirichlet Sampling (Dir) partition method (Hsu et al., 2020) to split the training set into $N^L$ subsets, each of which is stored in local clients as its local training data. After partitioning the whole training dataset into heterogeneous subsets, we sample a subset of half the classes as "Seen" categories in the original training set, and only local data from these classes are treated as potentially labeled data. In these potentially labeled data, 50% of instances of each labeled class are randomly sampled to form the labeled set. The remaining local data are treated as unlabeled data when training. We set $N^L = 5$ and partition the whole dataset into local data with $Dir(0.05)$ in our main experiments and explore more settings in Section 5.5 and Section 5.4.

**Evaluation Protocols.** Different from GCD who use the semi-supervised k-means to evaluate the final performance of models, we evaluate the global model performance with clustering accuracy (ACC) of prediction directly with k-means for all methods except for FedoSSL. We choose k-means because (1) it is more efficient in terms of running time; (2) differences between semi-supervised k-means and k-means have little effect on the comparison of different GCD methods. We follow the original evaluation method of FedoSSL due to its contribution related to its novel classifier calibration module. The cluster accuracy is defined as $ACC = \frac{1}{M} \sum_{i=1}^{M} \mathbb{1}(y_i, p^*(\hat{y}_i))$, where $y_i$ denotes the ground truth label, and $\hat{y}_i$ denotes the prediction given by k-means on extracted features, $p^*(\cdot)$ denotes the optimal permutation maximize the ACC of the overall predictions. Besides overall clustering accuracy testing on 'All' classes, we also measure the clustering accuracy for "Seen" and "Unseen" categories individually. Following common practice in GCD, we apply k-means on the feature outputted by [CLS] token of ViT to get the final clustering results.

**Baselines and implementation details.** We compare FedGCD-P with two strong baselines adapted from centralized GCD. GCD (Vaze et al., 2022) is the most classic algorithm designed for GCD problems, while simGCD (Wen et al., 2023b) is the state-of-the-art method for GCD in the centralized setting. Specifically, we utilize the original GCD methods in local client training to fine-tune the last block of ViT in both methods, and aggregate the trained local model with FedAvg (McMahan et al., 2017). We also compare FedGCD-P with existing FedGCD works FedoSSL (Zhang et al., 2023a) and AGCL (Pu et al., 2023a). As ACGL (Pu et al., 2023a) official implementation hasn't been released, we collect the baseline result of ACGL from (Pu et al., 2023a). We set the batch size to 128 for all methods and datasets. All methods are optimized by SGD with a momentum of 0.9 and weight decay of $1 \times 10^{-4}$ for 200 epochs with a cosine annealing schedule starting from a learning rate of 0.1. All methods use dino-vitb16 (Caron et al., 2021) as the backbone following the previous setting in GCD works (Vaze et al., 2022; Pu et al., 2023a). The hyper-parameters $\lambda$ controlling weight of supervised and self-supervised learning are set to 0.35 for Fed-GCD, Fed-SimGCD, and FedGCD-P. During training process, we define the initial random seed as 2023, then the random seed is increased by 1 every global round, while when federatedly partitioning the datasets, the random seed is set as 0. Our local training epoch is set as 1. Other parameters and implementation details follow the official implementation of baselines.

**Hardware and Software Configuration.** We conduct experiments using NVIDIA V100 32GB GPU, 20 Intel(R) Xeon(R) Silver 4114 CPUs @ 2.20GHz. The operating system is Oracle Linux 8 (x86_64) UEK Release 6. The pytorch version is 1.12.1. The numpy version is 2.0.1. The cuda version is 11.8. All our experiments are carried out in a single GPU to avoid unexpected influence on contrastive learning of splitting batch data to multiple GPUs.

## 5.2 Experiment Results

From Table 1, we observe that FedGCD-P significantly outperforms other baselines on the CIFAR-10 and CIFAR-100, while FedGCD-P ranked second in both CUB-200 and Oxford-Pet with slight with the best results. What's notable for the advantage of FedGCD-P is its significantly lower communication cost compared with other baselines. As shown in Table 2, FedGCD-P transmits only $0.70 \sim 0.78\%$ of parameter compared

with baselines. This advantage is in the requirement of real-world FL applications where many clients like edge devices may be short of communication bandwidth.

### 5.3 Ablation Study of FedGCD-P

We attempt to explain the effects of the components in our methods, with analysis towards different loss terms of FedGCD-P: (1) self-supervised contrastive loss(self-CL); (2) supervised contrastive loss (sup-CL); (3) self-distillation loss with entropy regularization (SD); (4) supervised classification loss (CLS); (5) prompt-tuning (PMT): If the column "PMT" is disabled, we only fine-tune the last block of ViT; while enabling "PMT" is to fine-tune [PMT] tokens (not include [CLS] token) instead of last block; (6) loss on prompt tokens (L-PMT). We have the following observation and conclusion: (1) imbalance between supervised and unsupervised learning has a catastrophic effect on FedGCD methods, which may lead to failure of convergence, e.g. (1) and (3) columns of Table 3, we believe this unique issue with FedGCD problems is associate with its decentralized setting and great heterogeneity make naive FedAvg aggregation unstable for the convergence of the global model, so we claim a more robust aggregation scheme is required to help more robust convergence of FedGCD methods, and believe it to be a promising future direction for FedGCD problems; (2) we show all components of our method FedGCD-P except for L-PMT is vital for achieving the best performance on FedGCD problems, especially the improvement brought by introducing prompt tuning, which we will experimentally verify later in Section 5.6.

Table 3: Ablation Study of FedGCD-P.

|  | self-CL | sup-CL | SD | CLS | PMT | L-PMT | CIFAR-100 (%) | | | Oxford-Pet (%) | | |
|---|---|---|---|---|---|---|---|---|---|---|---|---|
|  |  |  |  |  |  |  | All | Seen | Unseen | All | Seen | Unseen |
| (1) | ✓ | ✗ | ✗ | ✗ | ✗ | ✗ | 55.0 | 56.3 | 53.7 | 75.7 | 66.5 | 85.6 |
| (2) | ✓ | ✓ | ✗ | ✗ | ✗ | ✗ | 63.6 | 63.4 | 63.8 | 84.6 | 82.8 | 86.4 |
| (3) | ✓ | ✓ | ✓ | ✗ | ✗ | ✗ | 56.5 | 55.6 | 57.4 | 78.9 | 68.2 | 90.2 |
| (4) | ✓ | ✓ | ✓ | ✓ | ✗ | ✗ | 66.0 | 62.1 | 69.8 | 82.6 | 81.9 | 83.3 |
| (5) | ✓ | ✓ | ✓ | ✓ | ✓ | ✗ | 69.9 | 68.4 | 71.5 | 85.9 | 86.9 | 84.9 |
| (6) | ✓ | ✓ | ✓ | ✓ | ✓ | ✓ | 69.3 | 71.3 | 67.2 | 83.1 | 85.4 | 80.7 |
| (7) | ✓ | ✓ | ✗ | ✓ | ✓ | ✓ | 46.7 | 46.9 | 46.5 | 79.4 | 78.8 | 80.1 |

### 5.4 FedGCD-P under different degrees of heterogeneity

In this section, we discuss a significant problem in FL named data heterogeneity, and empirically show the robustness towards different degrees of heterogeneity of baselines and our method. From Table 4, we found besides the abnormal convergence of FedoSSL, all baselines suffered with the increase of heterogeneity from $Dir(0.2)$ to $Dir(0.05)$, especially Fed-SimGCD. The performance of FedGCD-P is at a medium level compared with other baselines.

### 5.5 FedGCD-P under different numbers of clients

In this section, we test the scalability of different methods in our FedGCD setting. We compare the performance of all methods under the FL system with 5 clients and 10 clients. In both settings, all clients participate in the local training and upload their local models for aggregation. From Table 5, we see the growth of the number of clients only slightly influences the performance of FedGCD-P and witness a relatively balanced decrease instead of in Fed-SimGCD, whose 'Seen' is increased while 'Unseen' greatly decreases. The phenomena show the supervised and unsupervised learning in FedGCD-P is relatively balanced. We attribute the good scalability to the alleviation of the overfitting problem in local data, which deteriorates with the increase of the number of clients and decrease of data samples on each client.

Table 4: Comparison of different degrees of heterogeneity Dir(0.2) and Dir(0.05), $N_L = 5$, on dataset CIFAR-100.*: we found FedoSSL cannot converge on CIFAR-10 and CIFAR-100 in our setting following their hyper-parameters and we attempt to rerun them with modified parameters but still cannot converge normally.

| Heterogeneity | Dir(0.2) | | | Dir(0.05) | | |
|---|---|---|---|---|---|---|
| | All | Seen | Unseen | All | Seen | Unseen |
| 5 clients (full participation) | | | | | | |
| **Fed-GCD** | 66.0 | 64.6 | 67.3 | 63.6 | 63.4 | 63.8 |
| **Fed-SimGCD** | 71.6 | 69.9 | 73.3 | 66.0 | 62.1 | 69.8 |
| **FedoSSL*** | 30.1 | 26.3 | 33.9 | 52.0 | 56.5 | 47.7 |
| **AGCL** | 56.1 | 56.8 | 55.3 | 54.2 | 54.6 | 54.0 |
| **FedGCD-P** | 71.7 | 72.6 | 70.8 | 69.3 | 71.3 | 67.2 |

Table 5: Comparison of different methods on $N_L = 5$ and $N_L = 10$, on dataset CIFAR-100, with Dir(0.05). *: we found FedoSSL cannot converge on CIFAR-100 in our setting following their hyper-parameters and we attempt to rerun them with modified parameters but still cannot converge normally.

| #clients | 5 | | | 10 | | |
|---|---|---|---|---|---|---|
| | All | Seen | Unseen | All | Seen | Unseen |
| **Fed-GCD** | 63.6 | 67.1 | 63.8 | 62.7 | 64.7 | 60.6 |
| **Fed-SimGCD** | 66.0 | 62.1 | 69.8 | 63.6 | 67.7 | 59.4 |
| **FedoSSL*** | 52.0 | 56.5 | 47.7 | 45.8 | 44.0 | 47.6 |
| **AGCL** | 54.2 | 54.6 | 54.0 | 52.2 | 53.6 | 52.4 |
| **FedGCD-P** | 69.3 | 71.3 | 67.2 | 69.0 | 71.1 | 66.9 |

### 5.6 Prompt can effectively alleviate overfitting

We empirically testify our motivation that prompt tuning can be utilized to alleviate the overfitting problem of local training. From Figure 3, other baselines' performances constantly decrease with the communication round after a short period of increase. It is clearly observed that FedGCD-P prevents the overfitting of the global model.

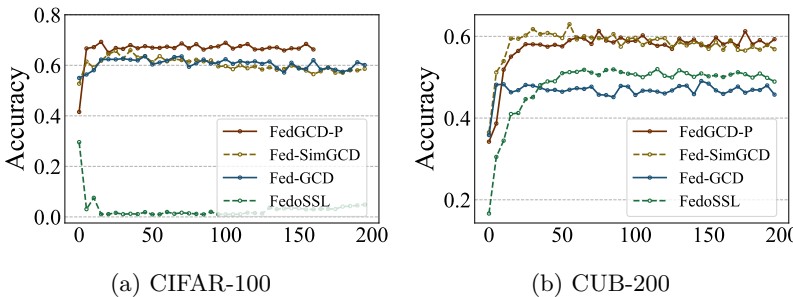

(a) CIFAR-100           (b) CUB-200

Figure 3: Performance of all methods on CIFAR-100 and CUB-200 with Dir(0.05), 5 clients.

## 6 Conclusion

In this paper, we introduce and formulate a federated generalized category discovery setting. We observe and dive into the common phenomena of overfitting problems in this FedGCD setting. We propose a global FedGCD framework based on a prompt tuning framework to solve the problem. We also show some

intermediate experiment results on both generalized and fine-grained datasets, and empirically testify our claim that we can significantly alleviate overfitting by fine-tuning prompt instead of ViT itself. For future work, we believe more advanced PEFT methods like lora can be incorporated into FedGCD-P to more effectively alleviate the overfitting problem due to deficient samples in local training.

**Acknowledgments**

LS and BH were supported by the NSFC General Program No. 62376235, CCF-Baidu Open Fund, HKBU Faculty Niche Research Areas No. RC-FNRA-IG/22-23/SCI/04, and HKBU CSD Departmental Incentive Scheme.

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
