# OpenReview forum: "Federated Generalized Novel Category Discovery with Prompts Tuning"
_TMLR — Accepted by TMLR_

### Review · Reviewer_MvSh · 2025-04-08

**Summary Of Contributions:**

This paper proposes a method for federated generalized novel category discovery (GCD). The paper identifies overfitting as one key observation when we apply ordinary GCD algorithms in the federated scenario. The authors propose a method that combines prompt tuning with existing GCD algorithms and apply it in FL, called FedGCD-P. Experiments show that the proposed FedGCD-P achieves higher accuracy and transmits considerably less data compared to existing baselines.

**Audience:**

Yes

**Broader Impact Concerns:**

No concerns.

**Claims And Evidence:**

No

**Requested Changes:**

1. Explain the phenomenon of overfitting in federated GCD in more details, e.g. what may be its contributing factors, what is the difference between GCD tasks and ordinary FL tasks, etc.
2. Explain the methodology more clearly (See weaknesses above).
3. Justify the wide applicability of the proposed FedGCD-P.
4. Provide more analysis on how PEFT addresses the overfitting issue of FedGCD.

**Strengths And Weaknesses:**

# Strengths
- The existence of novel categories in the FL setting is a practical observation. I think this is a practical problem but has received relatively few observations.
- The observation that prompt tuning addresses the overfitting issue is interesting, and, if better justified, will bring insights to the broader community.

# Weaknesses.
- The paper claims that 'discovering the overfitting problem of federated GCD' is one of its main contributions. However, in my opinion, the study into this phenomenon is weak. Figure 1 is the only figure that explains the observation, but the descriptions are very vague, and does not transmit sufficient insights to readers. Although I think that this observation will be of interest to a wider audience, I think it requires more efforts from the audience, e.g. how is it different from common overfitting phenomena in FL, what are the concrete observations, and what factors may be causing this issue.
- The paper applies prompt tuning technique for federated GCD, which may limit its practical applications. From my understanding, prompt tuning can only be applied on vision transformer models, but not on CNN models, which still plays a non-negligible role in vision recognition. Therefore, the authors may need to justify the wide applicability of the proposed FedGCD-P.
- The methodology is vaguely described. From my understanding, the methodology significantly relies on [CLS] and [PMT] tokens, but these relevant tokens are not described sufficiently in the paper, and it may be hard to readers to understand their meanings and their importance. In addition, the section 'loss on PMT token' is also very vaguely described. Finally, the self-supervised contrastive loss in Eqn. 4 is not easy to understand. Since it also involves the concept of $N(x_i)$, it seems not applicable to unlabeled data, which limits its practicality as a self-supervised loss.
- Experiments can also be improved. In my opinion, since 'overfitting' is a key concept and motivation of this paper, more efforts should be put into analyzing it. and the current Fig. 3 is insufficient. At least, the authors should show the improvement of training accuracy alongside the decrease of test accuracy. Some more possible experiments include, e.g. analyzing the gap between training & test, analyzing how the heterogeneity plays an important role in the train-test gap, and how the prompt tuning parameters (e.g. # tokens) impact the overfitting phenomenon.
- Minor points
    - In large language models, it is more or less known that full fine-tuning will lead to overfitting, while PEFT can mitigate it (e.g. Xu et al. 2023). The authors may need to justify how different is the task of GCD in terms of 'full fine-tuning leads to overfitting, while PEFT does not'.
    - In the Introduction, there is a broken reference (Appendix ??). The appendix is also nowhere to be found.


Xu et al. Parameter-Efficient Fine-Tuning Methods for Pretrained Language Models: A Critical Review and Assessment. Arxiv 2023

---

> ### Author Response · Authors · 2025-05-08
> **Responses for Reviewer #MvSh**
>
> We sincerely thank the reviewer for taking the time to review. According to your insightful comments, we provide detailed feedback below.
>
> ### Weakness "Minor points":
> 1. Please see below response to "Requested Change 1".
> 2. Sorry for our carefullessness. We had removed the ***Appendix*** section and move the contents forward but forget to change the corresponding reference. We have revised the reference to the correct position, i.e., ***Section 2.4***.
>
> ### Requested Change 1:
>
> We believe there exist two types of "overfitting problem" in FedGCD as stated in ***Second paragraph of Introduction***, i.e. (1)"a test performance gap between local training data and test data is witnessed due to the deficiency of training data", and (2)"overfitting for labeled (old) class is observed from a performance descent in a novel class". We believe these two kinds of overfitting are essentially bias toward the majority of data over the minority (here majority refers to old data for GCD problem, and data from classes with more samples on the client for FL problem) and regard prompt tuning as the solution for both of them.
>
> ### Requested Change 2:
>
> After careful checking, Eq. 4 in our paper doesn't involves the concept of $N(x_i)$. We guess your meaning is Eq.6 shouldn't have involved the concept of $N(x_i)$ as a self-supervised loss, we have revised Eq. 6 to its correct formula, which doesn't involves the concept of $N(x_i)$.
>
> ### Requested Change 3:
>
> First, for GCD problems, recent studies [1] [2] [3] seldom use CNN as its model architecture. Second, applying of prompt tuning technique is meant to design for the ViT architecture, there are plenty of PEFT methods for CNN like linear probing that can be used for mitigating the overfitting problem of CNN if it is used for GCD problem. Due to these two reason, we believe usage of prompt tuning will not hinder the practical application of FedGCD-P in GCD problem.
>
> ### Requested Change 4:
>
> The decrease of training loss (also seen as the improvement of training accuracy in most previous studies) alongside the decrease of test accuracy is shown in Fig. 1. We may need some time to carry out more experiments on how PEFT addresses the overfitting issue of FedGCD and present them in our revised version.
>
>
> > [1] Wen, Xin, Bingchen Zhao, and Xiaojuan Qi. "Parametric classification for generalized category discovery: A baseline study.", in ICCV 2023.
> >
> > [2] Pu, Nan, Wenjing Li, Xingyuan Ji, Yalan Qin, Nicu Sebe, and Zhun Zhong. "Federated generalized category discovery.", in CVPR 2024.
> >
> > [3] Zhang, Jie, Xiaosong Ma, Song Guo, and Wenchao Xu. "Towards unbiased training in federated open-world semi-supervised learning.", in ICML 2023.

---

### Review · Reviewer_MGkK · 2025-04-13

**Summary Of Contributions:**

This paper identifies the overfitting issue in Federated Generalized Category Discovery (FedGCD) tasks and proposes a prompt-based method to mitigate it. The proposed approach achieves performance improvements while reducing communication costs.

**Audience:**

Yes

**Broader Impact Concerns:**

None.

**Claims And Evidence:**

Yes

**Requested Changes:**

Please address the issues in the **Weaknesses** section.

**Strengths And Weaknesses:**

## **Strengths**

- The paper identifies and articulates the overfitting issue in FedGCD settings, which is a under-discussed problem.
- The proposed prompt-based method is experimentally validated and shows performance improvements.

---

## **Weaknesses**

1. **Typographical and Citation Issues**
   - There are several minor typographical errors that need attention. For instance, the phrase “as described in Appendix ??” lacks a proper reference and breaks the reading flow.

2. **Writing and Clarity**
   - Some sentences are overly complex and difficult to follow. For example, the sentence starting with *“Similar to GCD...”* on page 2 is long and nested—splitting it into shorter sentences would help clarity.
   - There are inconsistencies in notation, particularly in Sections 3.1 and 3.3. Label sets are referred to alternately as **$\mathcal{L}_n^L$** and **$Y_l$**, which can be confusing.
   - Font styles such as italic and calligraphic are used inconsistently (e.g., **$\mathcal{L}_I^L$** vs. **$L_n^L$** in Section 3.3). Please ensure consistent formatting throughout.

3. **Missing or Ambiguous Information**
   - Figure 1 lacks context. It’s unclear whether subfigure (a) illustrates standard GCD or the FedGCD setting. Providing this clarification, along with a brief description of the dataset used, would improve readability.
   - Figure 2 lists four loss types, but the ablation study in the text discusses six. The additional two are not clearly introduced in the method section, which could confuse readers.

4. **Potential Contradiction in Claims**
   - Section 4.1 states that overfitting is not observed in centralized settings, yet Figure 1 appears to suggest otherwise. The sentence *“Similar to GCD [29], we found the overfitting problem of GCD even worse and more complicated in the data-distributed scenario”* seems to imply overfitting is present in both. Please clarify the distinction more explicitly to avoid confusion.

5. **Unclear Purpose of Equations (5) and (6)**
   - Equations (5) and (6) are nearly identical, differing only by the dataset to which $x_i$ belongs. However, this distinction isn’t well explained. It would help to explicitly clarify why both are needed and what unique purpose each serves.

6. **Fairness of Baseline Comparisons**
   - The comparison with AGCL may not be fair. The reported performance of Fed-GCD in the AGCL paper (e.g., 80.7 on CIFAR-10) is significantly lower than what is reported here (90.3), suggesting possible differences in setup.
   - Either re-run AGCL under the same experimental configuration or adjust this paper’s settings to match AGCL’s to ensure a fair comparison.

7. **Ablation Study Concerns**
   - In the third ablation experiment, removing SD (a proposed module) actually improves performance. This casts doubt on whether SD is truly beneficial. Please discuss this result in more detail and justify the component’s inclusion.

8. **Questioning the Necessity of the Solution**
   - The proposed method reduces overfitting by essentially applying a lighter fine-tuning strategy. Could simpler techniques like early stopping, reducing local epochs, or lowering the learning rate achieve similar results? Please clarify why the prompt-based approach is preferable or uniquely effective.

9. **Generalization to FedGCD**
   - It’s unclear whether the proposed solution is specifically designed for FedGCD. Overfitting in ViTs is a common issue in FL settings in general. The paper would benefit from a clearer explanation of why this approach is particularly suited to FedGCD rather than FL more broadly.

10. **Motivation Could Be Stronger**
    - Assuming the core focus of the paper is to tackle overfitting in FedGCD, the motivation could be more explicitly and convincingly presented. Positioning overfitting as a **central challenge** and comparing the proposed solution to **existing mitigation strategies** would better highlight the novelty and necessity of this work.

---

> ### Author Response · Authors · 2025-05-08
> **Responses for Reviewer #MGkk**
>
> We sincerely thank the reviewer for taking the time to review. According to your valuable comments, we provide detailed feedback below and add them into our revision.
>
> ### Weakness 1:
> Thank you for pointing out the typos. We have re-checked our paper and revise the typos in our paper.
> ### Weakness 2:
> Thank you for your valuable suggestion that help us improve our writing. We have rewrite the complex sentences and potentially inconsistent notation and font styles you mentioned in our revision.
> ### Weakness 3:
> 1. We have added detailed explanation on subfigures and dataset in ***the caption of Figure 1*** in our revision.
> 2. "PMT" is not a part of loss, instead it refers to whether fine-tuning prompt or last layer of ViT. In ***the text part of Section 5.3***, we define "L-PMT" as "Loss on [PMT] tokens", which has been introduced in ***Loss on [PMT] tokens part of Section 4.1***.
> ### Weakness 4:
> Sorry for the unclear clarification on overfitting problem. First, the "Global Model" sub-figure shows that it is the performance of global model trained under federated setting instead of centralized setting. Then, acutally there exist two types of "overfitting problem" as stated right after the sentence “Similar to GCD [29], we found the overfitting problem of GCD even worse and more complicated in the data-distributed scenario”, i.e. "a test performance gap between local training data and test data is witnessed due to the deficiency of training data", and "overfitting for labeled (old) class is observed from a performance descent in a novel class". The presentation in Section 4.1 is meant to state the former is not observed in GCD. We revise the presentation in Section 4.1 to distinguish the two kinds of overfitting.
>
> ### Weakness 5:
> Sorry for the mistake in our footnote of second sum operation in Eq. (6). It should be revised to
> $$
> L_{self-con}(D_n^L)=\frac{1}{N_n^L} \sum_{x_i \in D_n^L} \sum_{x_i, x_i' \in \text{Aug}(D_n^L)\atop x_i \neq x_i'} \frac{\text{exp}(z_i, z_i')}{\sum_{x_n \in D_n^L} \text{exp}(z_i, z_{i}')}
> $$
>
> , similar to self-supervised classification loss in Eq. (4).
>
> The differences of Equation (5) and (6) is that it carries out contrast learning on different datasets, in Eq. (5), it carries out contrast learning by data label (classes), that is it is used to help distinguish data from different classes. While for Eq. (6), it helps model to separate all different samples in representation space.
>
> Thank you for your valuable reminder on mistakes of our equations.
>
> ### Weakness 6:
> Since official code of AGCL is still under construction, it would be difficult to re-run AGCL under the same experimental configuration. Moreover, some details like which is not complete in the paper of AGCL, thus it also unavailable to totally align our experimental setup to theirs.
>
> ### Weakness 7:
> This pheonomenon may be attributed to the pairing effectiveness of supervised loss and self-supervised loss, similar to the situation of ***line (1) of Table 3***, when only with self-CL, and without sup-CL, the performance is even worse than with SD. It may also attribute to the unbanlanced weight of supervised loss and unsupervised loss. The loss more largely depends on the unsupervised part, which leads to slower convergence on supervised part and thus worse performance. We are also carrying out experiment of removing SD component to verify the effectiveness of it.
>
>
> ### Weakness 8:
>
> Thank you for your suggestion on simpler methods. Below we analyze several ways of avoiding overfitting you mentioned.
>
> Early Stopping: we report our best accuracy during the training, therefore it already can be seen as a early stopping, we find early stopping may be useful but itself may not solve the problem substantially. We will also carried out experiment with early stopping techique during the rebuttal period.
>
> Reducing local epoch: As stated in ***Section 5.1 Baselines and implementation details*** in our revised pdf, our local epoch is set as 1.
>
> Lowering the learning rate: we have tried different learning rate and choose learning rate suitable for most baselines. Other learning rate either could not converge or got worse performances on most baselines.
>
> ### Weakness 9:
>
> We believe there exist two types of "overfitting problem" in FedGCD as stated in ***Second paragraph of Introduction***, i.e. (1)"a test performance gap between local training data and test data is witnessed due to the deficiency of training data", and (2)"overfitting for labeled (old) class is observed from a performance descent in a novel class". We regard prompt tuning as the solution for both the two kinds of overfitting. With the experiment results shown in our paper, we think prompt tuning may also has the potential to mitigate type(1) overfitting more broad FL setting.
>
> ### Weakness 10:
>
> Thank you for your advice. We have highlighted in ***Abstract*** to make our core focus on tackling overfitting in FedGCD more clear.

---

> > ### Comment · Reviewer_MGkK · 2025-05-08
> > **Additional Comments**
> >
> > Thank you for the detailed rebuttal and revisions. I appreciate the authors' effort in addressing the concerns. I have a few follow-up suggestions and questions for further clarification:
> >
> > 1. **Missing Definition of \( z' \) and Other Equation Clarifications:**
> >    In Equation (6) and related parts, the variable \( z' \) is used but never explicitly defined in the text. From the figures, I can only guess that it refers to an augmented or transformed view of the original image. Please consider adding a clear definition of \( z' \), and double-check other equations for potential ambiguities or missing terms.
> >
> > 2. **Implementation of “Loss on [PMT] Tokens”:**
> >    The explanation of how the “Loss on [PMT] tokens” is computed remains unclear. Is the loss applied to the prompt tokens in the same way as classification loss is applied to the [CLS] token (e.g., by attaching an MLP head)? A more concrete and implementation-specific description would be helpful for reproducibility and clarity.
> >
> > 3. **Distinction Between L-PMT and “(5) prompt” in Section 5.3:**
> >    In Section 5.3, it’s not fully clear how “Loss on prompt tokens (L-PMT)” differs from the setup in ablation line (5) where “prompt” is added. Does “prompt” in line (5) refer to standard prompt tuning, while lines (1)–(4) are purely fine-tuning without any prompt? Please clarify this distinction explicitly.
> >
> > 4. **Interpretation of Table 3 and Ablation Logic Order:**
> >    In Table 3, line (5) appears to outperform line (6), which might suggest that L-PMT brings limited or even negative contribution. If that is the case, please discuss why L-PMT is still necessary.
> >    Furthermore, the current order of ablation results could be confusing. A more intuitive ordering might be: start with classical components like CLS, then SD, followed by supervised contrastive loss (sup-CL), self-supervised contrastive loss (self-CL), and finally prompt-based variants. This sequence would better reflect the logical buildup of components—assuming the latter five are newly proposed in this paper.
> >
> >
> > 5. **Clarification of Contributions in Methodology (Section 4):**
> >    It would be helpful to explicitly state which losses in Section 4.1 are novel contributions of this paper, especially those not previously explored in the FedGCD literature. Highlighting this would enhance the clarity of your technical contributions and their relevance to the field.

---

> > > ### Author Response · Authors · 2025-05-09
> > >
> > > Dear Reviewer #MWcD,
> > >
> > > Thanks for your reply and new insightful questions. Please see our following feedbacks based on your new questions.
> > >
> > >
> > > **Q1:** Missing Definition of (z') and Other Equation Clarifications.
> > >
> > > ***Ans for Q1:***
> > > Thanks for your suggestion. We have added explanation of $z_i$, $z_j$, and $z_i'$.
> > >
> > >
> > > **Q2:** Implementation of “Loss on [PMT] Tokens”.
> > >
> > > ***Ans for Q2:*** Yes, we had explained how loss on [PMT] tokens are computed in ***Section 4.1 Loss on [PMT] tokens part***, "Therefore, we added auxiliary MLP for prompt tokens to assist the supervision from labeled samples and the self-supervision from all samples, our overall training objective follows the above overall learning objective in equation 7". We have highlighted this sentence for your reference and added Eq. (8) to better clarify the loss in our revision.
> > >
> > > **Q3:** Distinction Between L-PMT and “(5) prompt” in Section 5.3.
> > >
> > > ***Ans for Q3:*** Thank you for this insightful suggestion. If the column "PMT"  (prompt) is disabled, we only fine-tune the last block of ViT; while enabling "PMT"  (prompt) is to fine-tune [PMT] tokens (not include [CLS] token) instead of last block. Lines (1)–(4) only fine-tunes last block of ViT. Line (5) fine-tunes [PMT] token without applying loss directly on [PMT] tokens, instead it fine-tunes [PMT] tokens with loss on [CLS] token, in a similar way in [1].Line (6) directly apply loss on [PMT] tokens. We have supplied above explanation of column "PMT" in ***Section 5.3***.
> > >
> > > **Q4:** Interpretation of Table 3 and Ablation Logic Order.
> > >
> > > ***Ans for Q4:***
> > > 1. Actually in Section 5.3, we discuss the function of may not be necessary, it only acts as a inital attempt of our designs. However, it also do not too much harm to FedGCD-P performance, so we keep it in our method part.
> > > 2. We order the ablation components by its usage oder in centralized GCD problem. GCD [2] first uses unsupervised and supervised contrastive loss to enhance representation ability of ViT. Then, simGCD [2] utilize self-distillation loss and classification loss to more directly distill knowledge from supervision signal. Finally comes to our proposed prompt-tuning components.
> > >
> > > **Q5:** Clarification of Contributions in Methodology (Section 4).
> > >
> > > ***Ans for Q5:*** Only [3] implemented GCD in Federated Setting, i.e. self-CL and sup-CL is explored in FedGCD setting. We add the sentence "We first introduce supervised classification loss and self-distillation loss, together with prompt-tuning to FedGCD setting." to better highlight our technical contribution.
> > >
> > > > [1] Jia, Menglin, Luming Tang, Bor-Chun Chen, Claire Cardie, Serge Belongie, Bharath Hariharan, and Ser-Nam Lim. "Visual prompt tuning.", in ECCV 2022.
> > > >
> > > > [2] Wen, Xin, Bingchen Zhao, and Xiaojuan Qi. "Parametric classification for generalized category discovery: A baseline study.", in ICCV 2023.
> > > >
> > > > [3] Pu, Nan, Wenjing Li, Xingyuan Ji, Yalan Qin, Nicu Sebe, and Zhun Zhong. "Federated generalized category discovery.", in CVPR 2024.
> > >
> > >
> > > Best regards and thanks,
> > >
> > > Authors of #4531

---

### Review · Reviewer_hQ5H · 2025-04-23

**Summary Of Contributions:**

This paper addresses generalized category discovery (GCD) within the federated learning (FL) setting, where clients possess unlabeled data from unseen categories. The authors propose FedGCD-P, a framework utilizing prompt tuning. This method involves clients performing prompt learning locally and then aggregating these prompts globally, allowing for efficient knowledge sharing and improved performance on both seen and unseen categories, as demonstrated through experiments on various datasets.

**Audience:**

Yes

**Claims And Evidence:**

Yes

**Requested Changes:**

* Page 1 refers to an appendix. However, the paper does not have any appendices.

* The federated setting does not seem very realistic. All experiments are with only 5 - 10 clients (with a full participation ratio), which is very small for FL.

* The overfitting problem is something already known. The model can easily overfit wherever we have small datasets compared to the number of trainable parameters.

* Some of the training details, such as seeds, are missing.

**Strengths And Weaknesses:**

**Strengths**

* The problem is very interesting.

* Prior work section is comprehensive.

* Prompt tuning (in general, PEFT methods) reduces the computation cost in FL.

* The authors conduct thorough ablation experiments to highlight the contribution of each component,


**Weaknesses**

* The novelty is very limited. The paper uses prompt tuning and self-distillation to improve the performance and avoid overfitting.

* Prompt tuning is only one of the PEFT methods. The paper does not compare with other PEFT methods, such as LoRA.

* Is the unseen data really unseen? All the ViT models have already been trained on common datasets, so it is difficult to judge whether the improvements come from learning the unseen data or reducing the overfitting.

---

> ### Author Response · Authors · 2025-05-08
> **Responses for Reviewer #hQ5H**
>
> We thank the reviewer for taking time to review. In light of your insightful comments, we offer responses below.
>
> ### Weakness 1:
> Among our three primary contributions stated in ***Introduction***, we believe our first contribution (1) We are the first to identify the overfitting issue in federated generalized category discovery (FedGCD), which significantly hampers the performance of existing FedGCD methods and increases system overhead in federated systems, is of great significance for improving FedGCD methods especially in the time of large models. We also propose effective method FedGCD-P to mitigate this problem and verify its efficiency and effectiveness.
>
> ### Weakness 2:
> We use prompt tuning as a initial attempt to make improvement on current FedGCD methods and mitigate the overfitting problem. We agree there is potential to apply more PEFT methods in FedGCD methods for future works. However, our main focus is to find and try to solve the overfitting problem in FedGCD, instead of pushing the performance limit of FedGCD methods. Therefore, we haven't do ablation study on PEFT methods. But follow your suggestion, we will do experiments on other PEFT methods to discover the potential of them for FedGCD.
>
> ### Weakness 3:
> For GCD problems, the training is done in self-supervised style, i.e. without the assess of label in the whole pretraining process. The data used in self-supervised pretraining is not considered seen in GCD problem, as it doesn't have the supervision signal, i.e. label paired with the data. In our experiments, we follow the previous setting in GCD works[1] [2], adopting the self-supervised pretrained model DINO [3] as our pretrained model.
>
> ### Requested Change 1:
> Sorry for our carefullessness. We had removed the ***Appendix*** section and move the contents forward but forget to change the corresponding reference. We have revised the reference to the correct position, i.e., ***Section 2.4***. Thanks for pointing out this issue.
>
> ### Requested Change 2:
> For client number, due to its unique attribute, FedGCD Problem are difficult to simulate in larger number (like 100 in classic Federated Learning simulation), we actually follow them, Moreover, It already costs a lot time and resumption to simulate FedGCD in 5 and 10 client, more scaled experiments may be very costly and unaffordable. For example, we need 48 hours V100 32G GPU time to carry out the FedGCD pipeline even on small datasets like Cifar10 and Cifar100, let alone on ImageNet-100, the computing time extends to nearly 15 days.
>
> ### Requested Change 3:
> We actually do experiment in ImageNet-100, and we also observed similar or even worse overfitting problem (gap between seen and unseen classes acurracies) like in smaller datasets, e.g. Cifar-10, Cifar-100 and CUB-200, therefore, we think overfitting problem is not getting mitigated naturally as the data scaled up in GCD problem. We attribute this phoenomeon to the uniqueness of overfitting problem in GCD, as in GCD the overfitting problem is not caused by the scasity of training data, instead, it is the bias toward labelled data compared with unlabelled data, which will not be mitigated due to the scaling of training data.
>
> ### Requested Change 4:
> Thank you for your insightful suggestion that help us clarify our training process. We have added details about our trianing process including random seeds and local epoch, following the previous study, "During training process, we define the initial random seed as 2023, then the random seed is increased by 1 every global round, while when federatedly partitioning the datasets, the random seed is set as 0"; and local training epoch is set as 1.
>
> > [1] Vaze, Sagar, Kai Han, Andrea Vedaldi, and Andrew Zisserman. "Generalized category discovery.", in CVPR, 2022.
> >
> > [2] Pu, Nan, Wenjing Li, Xingyuan Ji, Yalan Qin, Nicu Sebe, and Zhun Zhong. "Federated generalized category discovery.", in CVPR, 2024.
> >
> > [3] Caron, Mathilde, Hugo Touvron, Ishan Misra, Hervé Jégou, Julien Mairal, Piotr Bojanowski, and Armand Joulin. "Emerging properties in self-supervised vision transformers." in ICCV 2021.

---

### Decision · Action_Editor_VHBd · 2025-06-04

**Recommendation:** Accept with minor revision

**Additional Comments:**

There is a lot of criticism regarding how the work is presented. I would like to particularly highlight the feedback from Reviewer MGkK who explained the many issues. The authors response, however, did not fully address it in my opinion. For instance, when the reviewer wrote "Please consider adding a clear definition of ( z' ), and double-check other equations for potential ambiguities or missing terms", the authors only responded by saying that fixed the definition of z'. I believe the authors need to make a thorough pass through the paper to fix any remaining issues. Ideally, they should also ask for feedback from their colleagues who are not familiar with the paper to identify any remaining parts that are not clear. Please note that there wouldn't be another round of reviewing, so I ask the authors to take this matter responsibly. The paper received mixed feedback, with two reviewers leaning reject, and I hope the authors take this information into account and acknowledge the need to improve the presentation.

Finally, since this response is not visible to the authors directly, I would like to provide the feedback from Reviewer MGkK' recommendation, so that the authors could better understand how their work can be improved:
> We acknowledge the authors' efforts in this work and appreciate the idea of using prompt finetuning to mitigate overfitting, which is a meaningful contribution. Therefore, we are inclined to give a Leaning Accept rating. However, the paper's writing quality appears to be suboptimal. It took considerable effort to fully understand the core ideas, particularly in the abstract and introduction sections. We strongly recommend the authors to reorganize the presentation of the problem and clearly articulate how the proposed method addresses it. Moreover, the current version of the paper does not fully reflect the thoughtful responses and clarifications made during the discussion phase. These should be properly integrated into the paper to improve clarity and overall quality. It would also be beneficial to incorporate feedback from other reviewers, as many of their suggestions are constructive and point toward potential improvements.

**Audience:**

Yes

**Audience Explanation:**

The paper identifies and articulates the overfitting issue in FedGCD settings, which is a under-discussed problem. All reviewers noted that the addressed problem is of interest.

**Claims And Evidence:**

Yes

**Claims Explanation:**

The authors identified a solid problem, provided a prompt-based method and experimentally validated its performance improvements. The reviewers noted that the related work was sufficiently covered, and the ablation experiments were thorough.

---

> ### Author Response · Authors · 2025-07-02
> **Revisions of our paper**
>
> Dear Action Editor VHBd,
>
> Thank you for your and three reviewers' valuable suggestions, according to which we have revised the presentation and correct mistakes presented in our work.
>
>
>
> ### Reviewer hQ5H:
>
> Weakness 1: We revised the summarization of our contributions to better demonstrate where our novelty lies in.
>
> Weakness 2: Follow reviewer's suggestion, we discuss the potential of combining FedGCD-P and PEFT methods other than prompt tuning, like lora.
>
> Weakness 3: We explain our experiment setup for choosing self-supervised pretrained model DINO.
>
> Requested Change 1: We have revised the reference to the correct position, i.e., ***Section 2.4***.
>
> Requested Change 4: We have added details about our trianing process including random seeds and local epoch in "Baselines and implementation details" part of ***Section 5.1***.
>
>
> ### Reviewer MGkK:
>
> Weakness 1: We have re-checked our paper and corrected the typos in our paper.
>
> Weakness 2: We have rewritten some difficult sentences to make them easier to read, such as *“Similar to GCD...”* on page 2, "On the other hand, cross-task transfer learning Transfer (TL) strategies..." and "However, transfer learning can do no help..." on page 1.
>
> Weakness 3: We have added explanations on Figure 1 and $L_{PMT}$ into our main text.
>
> Weakness 4: We have revised the presentation in both ***Introduction and Section 4.1*** to better distinguish the two kinds of overfitting.
>
> Weakness 5: We have re-checked the equations in ***Section 3 and 4*** in detailed.
>
> Weakness 7: We have carried out experiment of removing SD component and found the performance of FedGCD-P greatly dropped due to the removal of SD component, which proves its effectiveness. The experiment results have been included in Table 3.
>
> Weakness 10: We have highlighted in ***Abstract*** to make our core focus on tackling overfitting in FedGCD more clear.
>
>
> ### Reviewer MvSh:
>
> Requested Change 1: We have revised the presentation in both ***Introduction and Section 4.1*** to better distinguish the two kinds of overfitting and the originality of them.
>
> Requested Change 2: We have corrected the Eq. 6 and re-check the whole ***Section 3 and 4*** to avoid similar mistakes.